# DNA Methylation and Immune Memory Response

**DOI:** 10.3390/cells10112943

**Published:** 2021-10-29

**Authors:** Nathalia Noschang Mittelstaedt, André Luiz Becker, Deise Nascimento de Freitas, Rafael F. Zanin, Renato T. Stein, Ana Paula Duarte de Souza

**Affiliations:** 1Laboratory of Clinical and Experimental Immunology, Healthy and Life Science School, Pontifical Catholic University, PUCRS, Porto Alegre 90619-900, RS, Brazil; Nathalia.Noschang@edu.pucrs.br (N.N.M.); meuimuno@gmail.com (A.L.B.); deisynf@yahoo.com.br (D.N.d.F.); 2Department of Health and Human Development, La Salle University, Canoas 92010-000, RS, Brazil; rafaelzaninn@gmail.com; 3Infant Center, Medical School, Pontifical Catholic University, PUCRS, Porto Alegre 90619-900, RS, Brazil; rstein@pucrs.br

**Keywords:** B cells, T cells, epigenetic, DNA methylation, memory response

## Abstract

The generation of memory is a cardinal feature of the adaptive immune response, involving different factors in a complex process of cellular differentiation. This process is essential for protecting the second encounter with pathogens and is the mechanism by which vaccines work. Epigenetic changes play important roles in the regulation of cell differentiation events. There are three types of epigenetic regulation: DNA methylation, histone modification, and microRNA expression. One of these epigenetic changes, DNA methylation, occurs in cytosine residues, mainly in CpG dinucleotides. This brief review aimed to analyse the literature to verify the involvement of DNA methylation during memory T and B cell development. Several studies have highlighted the importance of the DNA methyltransferases, enzymes that catalyse the methylation of DNA, during memory differentiation, maintenance, and function. The methylation profile within different subsets of naïve activated and memory cells could be an interesting tool to help monitor immune memory response.

## 1. Introduction

Memory is a hallmark of adaptive immunity mediated by lymphocytes. The immune memory feature is the capacity of B and T cells to respond more effectively after a second encounter with the antigen [1,2]. In general, memory cells can persist in the absence of an antigen and in a higher number than naïve T cells of the same specificity. Understanding the generation of memory immune response is vital to help answer fundamental questions. For example, why some infections lead to protective immunity while others do not and why some vaccines are very effective while others are not.

Many mechanisms involving the generation of memory have been elucidated, among which epigenetic modifications stand out as the most recent one [3,4]. Conard Waddington introduced the term ‘epigenetic’ as a concept of environmental influence on inducing phenotype modification [5,6]. Epigenetics is the study of all hereditary and possibly reversible changes in the function of the genome that do not alter the sequence of nucleotides within the DNA. There are three primary types of epigenetic regulation: DNA methylation, histone modification, and non-coding RNAs expression, such as microRNAs (miRNAs) and long non-coding RNAs (lncRNAs). DNA methylation occurs by adding methyl group to the fifth carbon atom of cytosine residues in sites rich in cytosine and guanine, known as CpG islands. Histone modification regulates gene expression guided by changes, such as the acetylation, phosphorylation, and methylation of amino acids in histone terminal tails. Finally, microRNAs are small non-coding RNAs that bind to messenger RNA, repressing its translation.

Histone modifications are associated with immunological memory development. The chromatin marks on histone H3, H3K27ac, H3K4me1, and H3K4m3, involved with gene expression, were investigated during memory CD4 development. Genes that are induced in memory CD4 T cells but not naïve cells presented H3K4m3 histone modification [7]. Several studies highlighted the importance of histone modification during CD8 T cell memory differentiation [8,9,10,11]. Similar to CD4 T cells, the accomplishment of H3k4me3 modification is associated with the gene regulation of memory CD8 T cells [12]. Germinal centre formation, important for memory B cell maturation, is related to the gain of histone marks H3K4me1, H3K4m3, and H3ac [13].

Studies have been described wherein microRNAs have a role in memory differentiation [14,15,16]. For example, the microRNA miR-146a, miR 17∼92, and miR-155 influenced memory T cell differentiation [17,18,19]. The miR 17∼92 is also important for a proper B cell response mediated by follicular T cells (Tfh) [20]. Recently, it was shown that CD40 signal alters miRNA levels in B cells contributing to memory differentiation [21].

In this article, we provide a comprehensive description of DNA methylation, since it is one of the most studied epigenetic modifications, and its role in memory immune response based on a review of the literature.

## 2. DNA Methylation

DNA methylation is a well-maintained process in both prokaryotic and eukaryotic cells. In mammals, DNA methylation is based on the covalent transfer of a methyl group to the C-5 position of the cytosine ring of DNA in a CpG dinucleotide context. Thus, DNA methylation is a chemical modification, affecting cytosine residues, forming 5-methylcytosine (5 mC). Most DNA methylation is critical for normal development, playing an important role in several processes, including genomic imprinting [22] and the promotion of the alteration of the chromatin structure to more condensed forms. The methylation status of DNA is normally considered to be inversely correlated with transcriptional activity, with transcriptionally silent genes being highly methylated. It also affects DNA accessibility to transcription factors and RNA polymerases, leading to the modulation of gene expression [23]. However, the function of DNA methylation might be more complex than previously suggested. The mechanism by which DNA methylation affects gene expression is diverse. On the one hand, DNA methylation can actively block the binding of certain transcriptional factors (TFs) to the promoter, impairing transcription [24]. On the other hand, TFs can recognise methylated DNA, recruiting other TFs to remodel chromatin and activate transcription [25]. Hence, methylated DNA can be recognised by proteins, such as methyl-CpG domain-binding protein (MDBs), and recruit histone deacetylase (HDACs), driving a repressed chromatin environment [26]. DNA methylation is also observed in the CpG islands in the gene body and it is positively associated with gene expression [27,28,29].

DNA methylation is designed and maintained by the combined function of three active DNA methyltransferases: DNMT1, DNMT3A, and DNMT3B [30]. DNA methyltransferase adds a methyl in the CpG islands using S-adenosylmethionine (SAM) as substrate. DNMT1 is responsible for the maintenance of the methylation. Since it preferentially methylates the hemi-methylated DNA, it copies DNA methylation patterns to the daughter strands during DNA replication on cell division [31]. In contrast, DNMT3A and DNMT3B are known as de novo methyltransferases, so they have preference for unmethylated CpG dinucleotides and set the initial pattern of methyl groups in place on a DNA sequence. The DNMT3-like protein, DNMT3-L, is a third member of the DNMT3 family, presents an inactive catalytic activity, but assists de novo methyltransferases by increasing other DNMT3′s abilities to bind to methyl group donors [32]. DNA methylation is considered a stable process [33]; however, the demethylation process can be performed by activation-induced cytosine deaminase (AID) [34] and by the ten-eleven translocation proteins (TET) [35]. TET1 was the first member of the TET family to be identified, followed by TET2 and TET3 [36]. TET enzymes perform demethylation by the oxidative conversion of 5-mC. TET function is regulated by substrate accessibility and expression levels [37].

Methylation is an important process for mammalian development and cell differentiation. The methylation patterns during germ line development suffer a rapid demethylation of the paternal genome at fertilisation and over early embryonic progression [38]. The role of DNA methylation during cell differentiation, for example, is highlighted during different stages of haematopoiesis. Lymphoid progenitors are specifically methylated at myeloid transcription factor binding sites [39,40,41].

Epigenetics, mediated by DNA methylation, is an important mechanism for the regulation of the immune system. The epigenetic modifications have been reported to be important during an innate immune response [42], regulating B cell fate and function [43] and controlling T cell differentiation and memory response [44,45]. The acquisition of immune memory is the hallmark of a protective immune response. During this process, naïve T and B cells, upon exposure to antigens during primary infection, differentiate into memory cells, acquiring specialised functions that allow the organism to respond faster to a secondary infection with the same pathogen. The differentiation of memory T and B cells involves extensive epigenetic changes, including DNA methylation, that are necessary to initiate and maintain a hereditary gene expression programme [46].

## 3. DNA Methylation and Memory T Cells

### 3.1. CD4 T Cells

After an encounter with the cognate antigen presented on the surface of the dendritic cells, CD4 T helper cells differentiate into specialised effector cells driven by the cytokine milieu during activation. CD4 T cells are diverse and well-characterised, displaying functional phenotypes such as T helper Th1, Th2, Th17, Treg, and Tfh, and more recently, Th22 and Th9. These specialised subsets secrete different cytokines and work in accordance with different cells, helping them in several activities, such as antibody production by B cells, maintaining CD8 T cell response, and the regulation of macrophage function [47]. In the presence of IL-12, primed CD4 T cells produce interferon-gamma (IFN-γ), so they are called Th1 and are important in the context of intracellular pathogens. In contrast, naïve CD4 T cells primed in the presence of IL-4 are named Th2 and can produce IL-4, IL-5, IL-10, and IL-13. Th2 cells are important in fighting parasites and are involved in allergies. Different cytokine combinations can induce Th17 and these cells produce IL-17, which is essential to control bacteria and fungi infection, but can also be associated with autoimmunity. Regulatory T cells (Treg) are characterised by FOXP3 expression and are required to maintain self-tolerance. Treg can acquire regulatory lineage after maturation in the thymus or being induced by cytokines. Finally, follicular helper cells (Tfh) are specialised in helping B cells function, secreting IL-21.

Epigenetic markers are associated with CD4 T helper subsets; for example, classical Th1 and Th2 differentiation has been accompanied by the demethylation of interferon-gamma (Ifng) and Interleukin-4 (IL-4) genes, respectively [1,48]. The importance of epigenetic regulation and T cell subset differentiation has been extensively reviewed elsewhere [49,50]. Here, we focus on DNA methylation and its role on memory CD4 T cells.

Memory CD4 T cells are classified as central cells (TCM) and effector cells (TEM), according to their capability to proliferate and produce cytokine and migrate to different compartments in the body [51]. TEM cells display a greater capacity for producing T helper 1 or 2 effector cytokines compared with TCM. However, studies have shown that TCM/TEM markers are insufficient to describe the potential function of memory CD4 T cells according to their different phenotypes [27,28,52,53]. The current model predicts that the qualitative composition of the CD4 T cell memory pool is likely determined during the primary response; the progeny of one CD4 T cell can give rise to both effectors and Tfh/TCM precursors [51,52,54]. Different factors influence CD4 T cell memory development, including MHC-II and T cell receptor (TCR) interaction [55], co-stimulatory signals from CD40, CD86, and OX40L on the antigen-presenting cell [56,57], and inducible costimulator-ligand (ICOSL) expression in B cells [52], cytokines (especially IL-2 [58,59]), and chemokine receptor CXCR3 expression in CD4 T cells [58,60].

The role of DNA methylation during memory CD4 T cell differentiation and function has been described by different groups in mice and humans. Studies have demonstrated that naïve, effector, and memory cells present unique characteristics in their methylation status, suggesting that the analysis of DNA methylation could be used as a marker to evaluate memory differentiation and commitment.

For example, Hashimoto et al. performed genome-wide DNA methylation analysis of murine T cells to elucidate the molecular signature of memory CD4 T cells. They compared naïve, effector, and memory CD4 T cells and found 1144 differentially methylated regions (DMRs) across the murine genome during the process of CD4 T cell differentiation. Interestingly, most of these methylated regions were in introns [61]. These DMRs were found in genes associated with cytokine production, homing to bone marrow, and immune responses. Methylation changes in memory CD4 T cells exposed to specific antigens appeared to regulate enhancer activity. Methylation profiles differed between memory T cell subsets, demonstrating a link between T cell methylation status and CD4 T cell differentiation, confirming that methylation can be used as a marker for identified different phenotypes [61].

DNA methylation is important for memory CD4 T cells to remember their previous effector lineage after antigen clearance and reacquire their lineage-specific effector functions upon antigen re-encounter. This concept was demonstrated in mice, showing that the granzyme b (Gzmb) gene is unmethylated in Th1 effector cells and remains unmethylated in Th1 memory cells. In contrast, the Gzmb gene in memory Tfh retains the naïve methylation programme. Th1 produce granzyme b, while Thf do not. The methylation of Gzmb can be used as a marker to distinguish CD4 Th1 cells from Tfh. Interlekin 21 (Il-21) and interferon-gamma (Ifng) genes are demethylated in both Tfh and Th1 and remain unmethylated in memory T cells. Consequently, Il21 and Ifng loci methylation status cannot be used to distinguish memory CD4 Th1 from Tfh. Those loci may be similarly poised for potential rapid re-expression in both subset of memory cells. [53]. More evidence that memory T cells can remember their previous effector phenotype was shown in mice, where Treg was shown to transiently downregulate yet retain a memory of Forkhead box protein (FOXP3) gene expression. Foxp3 is a crucial regulator of Treg gene expression, and the demethylated status of the Treg cell-specific demethylation region (TSDR) appears to provide the basis of such an epigenetic memory of FOXP3. Memory Foxp3 ^-^CD44^hi^CD4^+^ T cells present a TSDR partially demethylated that after antigen stimulation can either express Foxp3 again, exhibiting a fully demethylated TSDR, or continue to be Foxp3-negative and exhibit a fully methylated TSDR [62] (Figure 1).

The control of DNA methylation during CD4 T cell memory differentiation and function is confirmed in human cells. In human CD4 T cells, the loss of methylation at the Ifng promoter at a conserved element termed CNS-1, located 4.2 kb upstream of the human *Ifng* gene, is associated with the development of functional IFN-γ in memory [63]. Additionally, in human cells, the transcriptional factor CREM-α mediates the epigenetic remodelling of the cytokines IL-2 and IL-17A genes during CD4 T cell differentiation, favouring memory effector cells. DNMT3A is important in the control of IL-2 and IL-17A expression in memory CD4 T cells [64]. Human memory Treg but not naïve Treg can produce IL-17A due to the methylation of the RORC locus, which controls IL-17A expression [65]. Additionally, DNA methylation is important in memory phenotype maintenance, since CD4^+^FOXP3^+^ cells derived from the in vitro stimulation of human CD62L^+^ central memory was more methylated in the FOXP3 TSDR compared to Treg, which could explain why these CD4^+^FOXP3^+^ did not exhibit suppressive function [66].

Using deep sequencing to trace the epigenetic landscape of human CD4 memory T cells, Komori et al. examined the methylation of 2100 genes and found that 132 genes are differentially methylated between naïve and memory T cells. They found that 21 genes exhibited differential methylation between naïve and memory CD4 T cells that correlated with differential expression of genes before activation, and 84 genes demonstrated differential methylation between memory cells and naïve cells that correlated with differential gene expression after activation [67]. Complementary to that, Durek et al. performed an in-depth analysis of the epigenome of human memory T cells, and they found that during memory differentiation, there is a progressive loss of DNA methylation in heterochromatin parts of the genome [68]. The entire genome methylation dropped from 84% in naïve CD4 T cells to 67% in terminally differentiated memory CD4 T cells. They also showed that the transcriptional factor FOXP1 is an important gatekeeper for the naïve to memory transition, and the promoter of FOXP1 presented a methylation-sensitive activity. The FOXP1 gene locus presented a DMR which displayed a strong progressive gain of methylation during CD4 T cell memory differentiation [68].

The importance of DNA methylation was demonstrated in specialised functions of memory CD4 T cells; the gene MYC is methylated and suppressed in cells that maintain autophagic memory [69]. Additionally, DNA methylation is important in gut homing receptor α4β7 expression on memory CD4 T cells. DNA methylation at the specific regulatory region of the Integrin-alpha 4 (Itga4) locus correlates with α4β7 receptor expressions and memory differentiation in mice. In humans, the Itga4 locus presents similar patterns of DNA methylation [70].

DNA methylation is also important in CD4 memory T cell function in the context of infectious diseases. In chronic HIV-infected individuals, memory CD4+ T cells with a terminally differentiated phenotype had increased IL-2 promoter methylation relative to healthy individuals [71].

### 3.2. CD8 T Cells

The response mediated by CD8 T cells is important for immunity against intracellular pathogens and tumours. During an infection or vaccination, naïve CD8 T cells are activated, undergo clonal expansion, and differentiate into effector cells. The effector function of CD8 associates with granzyme and perforin production to perform the cytotoxic activity. Additionally, these cells produce important cytokines, such as interferon-gamma (IFN-γ), which moderate cell-mediated immunity. The CD8 T cell differentiation programme starts after dendritic cell antigen presentation (signal 1), co-stimulation (signal 2), and inflammation (signal 3) [72]. This programme is accompanied by robust proliferation and transcriptional, epigenetic, and metabolic changes. The effector cells are either terminally differentiated or cells that retain the potential (memory precursor effector cells—MPECs) to further develop the long-lived memory CD8 T cells. MPECs can be distinguished based on their expression of CD127 (Interleukin 7 (IL-7) receptor alpha) and decreased expression of KLRG1 (killer cell lectin-like receptor G1) [73]. Additionally, some cells expressing higher levels of KLRG1 can downregulate it and differentiate into memory cells’ T cell lineages [74]. If there is persistent antigen exposure, CD8 T cells can be induced to a terminal non-functional state of differentiation called exhaustion [75].

The majority of effector CD8 T cells will die during a contraction phase and the remainder that survive (around 5%) will give rise to the pool of long-lived memory CD8 T cells. Several transcriptional factors are described to control memory CD8 T cell differentiation and function, including T-box 21 proteins, Eomesodermin (Eomes), B-lymphocyte-induced maturation protein 1 (BLIMP-1), T cell factor 1 (TCF1), ID proteins [76,77,78,79,80], Signal transducer and activator of transcription 3 (STAT3), and interferon regulatory factor 4 (IRF4) [81].

As with CD4 T cells, there are different types of memory CD8 T cells, such as effector memory T cells (TEM), central memory T cells (TCM), stem-cell memory T cells (TSCM), and resident memory T cells (TRM). They are different in their homing capacity, multipotency, and cytotoxic function. In addition, some studies indicate the presence of virtual and innate memory CD8 T cells subsets, which developed in uninfected individuals and acquire phenotypic markers and functional properties similar to memory cells [82,83,84].

The exposure to effector-promoting signals allowed memory CD8 T cells to recall effector functions while retaining the naïve-like capacity to develop into other memory and effector cell types, and these features are associated with epigenetic mechanisms [85,86,87,88]. Memory CD8 T cells can be considered an assembly between naïve and effector cells regarding their epigenomic profile [8]. The importance of DNA methylation in the development and maintenance of memory CD8 T cells has been extensively studied in mice and humans.

The methylation profiling of effector versus memory-precursor CD8 T cells provided evidence that effector CD8 T cells can dedifferentiate into memory T cells, acquiring de novo DNA methylation programmes at naïve associated genes [85]. Additionally, the DNA methylation status of memory CD8 T cells from mice repetitively infected with Listeria monocytogenes is different between primary and secondary responses, increasing the unmethylated regions in memory CD8 T cells during secondary response. The differences in methylation profile are mostly in the distal regions from transcription start sites, which include transcriptional enhancers [89].

The role of the enzymes associated with methylation and demethylation processes during memory CD8 T cell differentiation have been described by some studies. The absence of TET2 (Tet methylcytosine dioxygenase 2) promotes memory CD8 T cell differentiation in mice. T cell receptor signalling increases the expression of TET2 mRNA in a Ca2+-dependent manner. TET2 depletion increases the formation of antigen-specific CD8 memory cells during a viral infection. These memory cells are effective in controlling the pathogen after challenge and expressing the transcriptional factor associated with memory differentiation, Eomes. The loss of TET2 leads to the hypermethylation of several regions that control memory fate [90]. Additionally, there was an increase in the proportion of CD8 memory precursor cells CD127^+^KLRG1^-^ in DNMT3A knockout (KO) mice infected with lymphocytic Choriomeningitis virus (LCMV), or influenza [91]. Similar findings were demonstrated by Youngblood et al., showing that DNMT3A KO mice have an effector CD8 T cell response to infection, similar to wild-type mice but displaying a greater formation of memory precursors [86].

Some studies found that methylation is important to control the expression of genes responsible for memory CD8 T cell function. One example is the expression of the programmed cell death protein 1 (PD1), a T cell inhibitory receptor. An adapted programme for the regulation of PD1 expression, with a re-methylation of the Pdcd1 locus, occurs during differentiation into functional memory CD8 T cells [92]. Additionally, the Ifitm3 gene promoter, which regulates the expression of interferon-induced transmembrane protein 3 (IFITM3), that confers broad resistance to viral infection, was selectively unmethylated in memory CD8 T cells in the lung during influenza infection in mice [93]. Another example is the expression of IFN-γ in memory CD8 T cells. One study demonstrated that IFN-g expression is regulated by DNA methylation by progesterone. Progesterone impairs the production of IFN-γ by memory CD8 T cells, inducing hypermethylation in the Ifng gene promoter region. This mechanism is suggested to be responsible for the susceptibility of pregnant women to Listeria monocytogenes infection [94]. An additional study showed that the IFN-γ production by virtual-memory CD8 T cells in nonimmune mice is regulated by DNA methylation. These memory-like cells presented lower DNA methylation rates at different CpG sites in the Ifng gene [95].

The importance of DNA methylation in memory CD8 T cell differentiation was confirmed in human cells. Rodriguez et al. performed DNA methylation profiling of naïve CD8 T cells, effector memory TEM, and terminally differentiated effector memory (TEMRA) [96]. Analysing the methylation pattern of the subpopulations, the authors noticed that in naïve cells, it was very different from TEM and TEMRA. TEM and TEMRA have a very similar methylation profile. They found that TEM differentiation was primarily associated with a global loss of methylation. Reviewing the function of differently methylated genes between naïve and memory cells, the authors found EOMES, T-box 21, and BLIMP1, which are key transcription factors, genes involved in cytotoxicity, such as perforin (prf1), IFN-γ, granzyme (gzmb, gzmh, gzmk), cytoskeleton (actn1,2,4, dock2, nuak2, coro2b, ssh1), cell adhesion (itgb1, cd58, itga6), and chemokine signalling (cxcr1, cx3cr1, cxcr4). They highlighted that DNA methylation contributes to the tuning of T cell receptor (TCR) signalling in resting TEM cells, since many important genes of the TCR signalling pathway go through de novo methylation. Additionally, they found DNA methylation changes during the memory differentiation in the Wnt/β-catenin and the TGF-β signalling pathway [96].

Akondy et al. demonstrated the importance of DNA methylation in human memory CD8 T cells through a longitudinal analysis of individuals vaccinated with a yellow fever vaccine [97]. Memory CD8 T cells presented more demethylated CpG sites near to the transcriptional region of the granzyme b gene compared to naïve cells. Additionally, the CpG region near the perforin gene was demethylated in memory cells. Memory CD8 T cells retained the epigenetic profile to permissive granzyme B and perforin expression [97] (Figure 2).

DNA methylation is also important in the control of the expression of CX3CR1 in effector memory CD8 T cells. CX3CR1 is the chemokine receptor for fractalkine, responsible for T cell migration to inflammation sites. Effector memory CD8 T had an increased expression of CX3CR1 combined with a decreased DNA methylation of its promoter gene. IL-7Rαlow TEM CD8 T cells experienced an increase in migratory capacity for the CX3CR1 fractalkine ligand compared to IL-7Rαhigh TEM CD8+ T cells, suggesting an important biological outcome of the differential expression of CX3CR1 [98] (Figure 2).

Studies have demonstrated that pathological conditions can change the methylation profile of memory CD8 T cells. Resident memory CD8 T cells from urinary bladder cancer patients had minor methylation of the perforin gene transcription enhancer region compared to memory CD8 T cells obtained from healthy donors [99]. In addition, Abdelsamed et al. (2020), seeking to better define the DNA methylation programmes associated with the developmental status of self-reactive CD8 T cells in type 1 diabetes, used whole-genome bisulphite sequencing (WGBS) analysis of the differentiated status of CD8 T cells [100]. CD8 T cells specific against beta-cells display a DNA methylation profile of memory stem-cells, showing a T cell pluripotency index based on DNA methylation to preserve autoreactivity ability. The article provided additional documentation that the gradual differentiation of human T cells is related to changes in DNA methylation [100].

## 4. DNA Methylation and Memory B Cells

B cells are characterised by the ability to mediate the humoral immune response due to differentiation into antibody-secreting cells through the activation of B cell receptors (BCRs). The humoral immunological memory is dependent on the acquisition of long-lived plasma cells that produce protective antibodies and memory B cells that can respond to reinfection [2,101]. During primary response, the generation of long-lived plasma cells and memory B cells from the naïve B cell repertoire occurs. The predominant response takes place on the secondary lymphoid organs in the B cell follicles and the germinal centres (GC) [102,103]. This complex process occurs in two phases. Phase 1 is characterised by the differentiation of naïve B cells into short-lived plasma cells and GC B cells in B cell follicles. Phase 2 is characterised by the differentiation of GC B cells into long-lived plasma cells and memory B cells [104]. During phase 2, GC B cells proliferate and undergo to somatic hypermutation to produce high-affinity antibody. GC B cells can differentiate into long-lived plasma cells that migrate to bone marrow and secrete large quantities of antibodies that persist for years. Additionally, GC can differentiate into long-lived memory B cells, which are quiescent. In a secondary encounter with an antigen, memory B cells respond by differentiating into long-lived plasma cells or by proceeding with the GC reaction again [2]. Both phases occurred with interactions between B cells and T follicular helper cells (Tfh cells) for a T-cell-dependent humoral response [105]. Memory B cells can also be induced by a T-cell-independent mechanism, previously reviewed by Defrance et al. The differences between them are associated with somatic hypermutation and isotype switching [106].

Memory B cells proliferate and differentiate more efficiently than naïve B cells upon antigen-specific or polyclonal stimulation [107,108]. Cells are long-lived and can serve to protect the host against re-exposure or to help the clear persistent primary infection. Epigenetic modifications are crucial for B cell differentiation and function [43]. Barwick et al. demonstrated that the genetic deletion of the de novo DNA methyltransferases genes, DNMT3a and DNMT3b (DNMT3-deficient), in B cells leads to normal B cell development and maturation [109]. However, DNMT3a gene depletion increased cell activation and the expansion of GC B cells and plasma cell populations upon immunisation in the mouse model [109]. Additionally, in human DNA, methylation is present in the early and late stages of B cell development [110].

The DNA methylome and transcriptome of human B cell subsets reveal the importance of DNA methylation for humoral response before and after antigen exposure in vivo. In a seminal study, Lai et al. described that in human tonsillar B cell subsets, memory B cells and plasma cells have different expressions of DNMTs to naïve cells after antigen exposure in vivo. GC B cells present elevated DNMT1 and DNMT3B expression and minimal expression of DNMT3A [111]. Naïve B cells have opposite DNMT expression in relation to GC B cells. The alteration of DNMT2 upon immune activation can be possible due to the reprogramming of the DNA methylation of GC B cells. DNA methylation changes in the GC are acquired by memory B cells and plasma cells [111]. In addition, they found that memory B cells and plasma cells share a similar methylome but have a distinct transcriptional programme. Memory B cells and naïve B cells presented a different methylome but a very similar transcriptional programme. They showed that during B cell differentiation, Alu elements are demethylated and accompanied by the repression of the DNMT3a gene. Alu elements are transposable elements which are very abundant in the human genome. The authors proposed that the loss of DNA methylation during naïve to GC B cell differentiation allowed the potential to generate plasma cell or memory cell fate [111] (Figure 3).

There are various intrinsic mechanisms involved in the protection of the long-term survival of memory B cells, such as autophagy [27,112]. Memory cells express higher levels of autophagy genes than naïve and GC B cells [113]. Autophagy can be regulated at the epigenetic level by DNA methylation in autophagy genes. In mice, epigenetic changes in DNA methylation do not induce the regulation of autophagy gene expression in memory B cells, but the transcription factors, such as FOXO1 and FOXO3, are necessary to control autophagy gene expression [114].

DNA methylation in B cells also has a role during pathology association with the immune response; one example is common variable immunodeficiency (CVID). CVID is characterised by the loss of B cell function. Rodriguez-Cortez et al. performed a comparison of CVID patients and healthy individuals, which revealed significant changes in DNA methylation associated with CVID in B cells, specifically the hypermethylation of several genes of relevance in B cell biology, including pik3cd, bcl2l1, rps6kb2, tcf3, and kcnn4 [115]. The memory B cells in a CVID patient cohort presented a prejudiced ability to demethylate and upregulate these genes in transitioning from naïve to memory cells. The authors suggested an impaired epigenetic signature in memory cells in CVID individuals, which can be related to decreased survival in CVID patients, and propose potential targets for clinical intervention [115].

Additionally, the role of DNA methylation could be noticed in patients with a rare immunodeficiency disorder, which presents centromere instability and facial anomalies syndrome due to hypomorphic mutations in DNA methyltransferase DNMT3B and DNA methylation that compromises the immune response [116]. These patients presented altered transcription, which affected disease-relevant genes, such as the memory B cell marker CD27 [116]. The role of DNA demethylation was also described in hyper-IgM syndrome patients. The absence of AID catalytic activity on these patients affected DNA methylation in naïve and memory B cells. The enzyme response for these active demethylation processes during the transition to memory is the TET [117].

## 5. Concluding Remarks

Significant progress has been made in dissecting the mechanisms of memory immune response differentiation and function and its association with DNA methylation. In this regard, the functions of enzymes that perform DNA methylation or demethylation have been studied during memory differentiation (Table 1). The role of DNMT3A is the most well-characterised enzyme in memory immune response. DNMT3A controls cytokine production in memory CD4 T cells [37,64], and the depletion of DNMT3A is associated with increased memory CD8 T cell differentiation [86,91] and increased CG B cells and plasma cells [109].

CD4 T cells present different functional phenotypes and DNA methylation regulation contributes to memory CD4 T cells, reacquiring their lineage-specific effector functions during secondary response. The regulation of DNA methylation is also essential to memory CD8 T cell differentiation and to the control of effector functions. Although there are fewer studies regarding DNA methylation in B cells compared to T cells, it is becoming clear that this process also has a role in memory B cell differentiation.

Some studies already investigated the methylome of human memory T and B cells and found differences in DNA methylation, which can distinguish memory cells from naïve and effector cells. Several genes are methylated and demethylated during memory generation. The methylation profile of memory cells could be a future tool to help monitor memory response during infection, immunotherapies, and vaccines.

## Figures and Tables

**Figure 1 cells-10-02943-f001:**
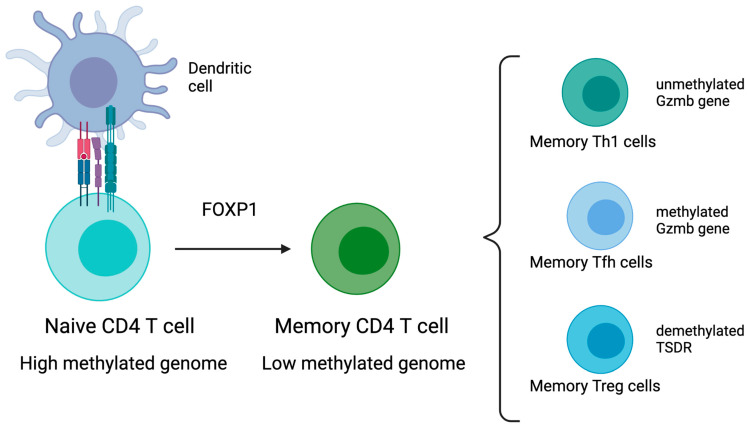
DNA methylation and memory CD4 T cell: memory CD4 T cells are generated following activation of naïve T cell by dendritic cell presenting peptide on major histocompatibility complex (MHC) molecules. FOXP1 is a transcriptional factor that controls memory differentiation and is regulated by DNA methylation. There is a progressive loss of DNA methylation during memory differentiation. DNA methylation of Gzmb gene can be used as a marker to distinguish memory CD4 Th1 from Tfh. The expression of Foxp3 in memory Treg is controlled by DNA methylation at TSDR (Treg cell-specific demethylation region).

**Figure 2 cells-10-02943-f002:**
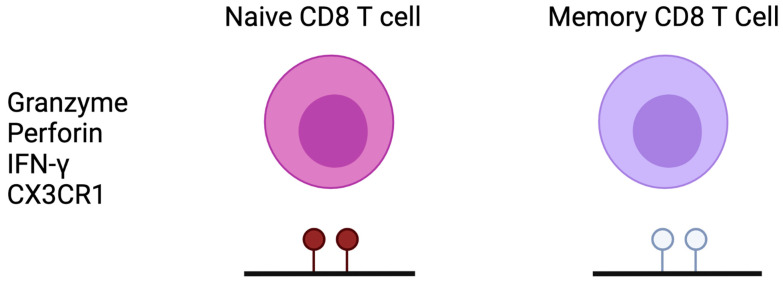
DNA methylation and memory CD8 T cell: memory CD8 T cells presented a decreased DNA methylation of granzyme, perforin, IFN-gamm and CX3CR1 genes.

**Figure 3 cells-10-02943-f003:**
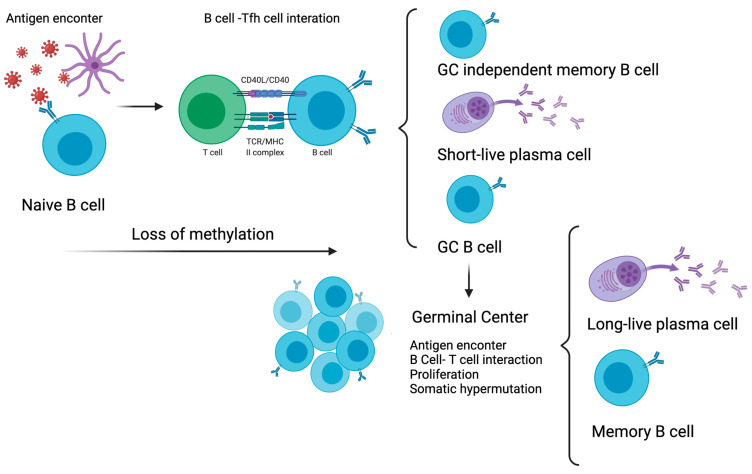
DNA methylation and memory B cell: A—differentiation of naïve B cells into short-lived plasma cells and germinal centre (GC) B cells. B—differentiation of GC B cells into long-lived plasma cells and secondary memory B cells. This process occurs in lymphoid organs in the B cell follicles and the germinal centres. The transition from naïve to GC B cell is dominated by loss of methylation in immune activation-induced DNA methylation regions.

**Table 1 cells-10-02943-t001:** Role of the enzymes that regulate DNA methylation during memory T cell differentiation.

	Enzyme Responsible for DNA Methylation	Enzyme Responsible for DNA Demethylation
Memory CD4 T cells	De novo DNA methyltransferase 3a (DNMT3a) interacts with CREM, mediating the epigenetic remodelling of IL2 and IL17A during memory CD4 T cell differentiation [64].	-
Memory CD8 T cells	The absence of DNMT3a promotes memory CD8 T cell differentiation in mice [86,91].	The absence of ten-eleven translocation (TET)2 promotes memory CD8 T cell differentiation in mice [90].
Memory B cells	DNMT3a expression is reduced in activated CG cells but is similar between naïve and memory B cells [111].	TET mediates the demethylation during the transition from naïve to human memory B cells [117].

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
