# Peer review of "DNA Methylation and Immune Memory Response"

_cells, 2021, doi:10.3390/cells10112943_

Round 1

Reviewer 1 Report

Dear authors,

I’ve read your manuscript “DNA methylation and immune memory response” where you remark the importance and relevance of DNA methylation in T and B cells. I’ve found it interesting.

It is well written with an nice rhythm to follow although a bit condensed in some paragraphs, missing also some other valuable reference. Please add them when need it.

In fact, in the introduction between lines 39-46 you listed the primary types of genetic regulation but nothing more is said upon histone modification and microRNAs. It is clear that you want to focus on DNA methylation but it is quite important to expand and explain more about the other two types, as they are relevant too in immunity, providing relevant references as well. Please add it in this section.

Furthermore, in line 98-100 you wrote that Th1 and Th2 differentiation is influenced by methylation of IL-4 and IFNG genes but it is not clear which gene for which subpopulation. It can be misleading, please rephrase it.

In line 112-115 you forgot to mentioned the costimulatory molecules expressed by APC cells, while you just mentioned ICOS of B cells. Please add it along with references.

In line 219-220 you wrote “the DNA methylation status in repetitively infected memory CD8 T cells (..)”. As it is written, the sentence suggests CD8 T cells being repetitively infected, while – I suppose- you want to say that CD8 T cells show different methylome in repetitively infected individuals. Please rephrase it.

In lines 295-305 you have described the two phases of B cells differentiation. I found it confusing to read as it is too brief. Please improve it as it is relevant for the paragraph.

In table 1 you have not mentioned the protein AID that can reverse the methylation although you have quoted it between lines 76-78. It is not involved in any processes of the T and B cells functionality?

As last, I would suggest to spell the entire names when proteins are mentioned all over the manuscript, followed by the acronyms.  It has not been done properly.

Reviewer 2 Report

The manuscript by Mittelstaedt at al. is a focused review summarizing the role of DNA methylation in the regulation of development of memory T and B cells. The review is detailed, logically organised and well-written, but quite specialised and not easy to follow for a non-expert reader. Inclusion of figures to illustrate the concepts would greatly improve understanding.

My main comment is that while numerous relevant research papers and concrete examples are enumerated to associate DNA methylation changes with memory cells, the concepts are not explained in enough details, as the mechanistic details are frequently completely missing. For example, the authors often mention that DNA methylation or a given methyltransferase is important for regulation of a given gene or process, but the mechanism of this regulation is not explained. It would be beneficial to mention whether (and how) the epigenetic changes are linked to expression changes and the biological function of changed genes/pathways. In addition, it is unclear whether the methylation changes are just associated with given phenotypes / cell subsets or may be driving them. When key transcription factors are mentioned, it would be useful to explain how their changed expression (or binding) may be regulating cell function / specialisation to better link the epigenetic change to biology.

Specific comments:

  • Several parts are confusing and/or not fully correct:
  • Line 41: “in regions rich in cytosine and guanine known as CpG” -> this is confusing, CpG are sites and not regions, and regions rich in cytosine and guanine are CpG islands, which are often unmethylated (unless in tissue specific or disease context).
  • Lines 52-53: “It suppresses ….” Change in chromatin structure is one of the mechanisms of how DNA methylation represses gene expression, so I would suggest reversing the order of arguments in that sentence
  • Lines 53-54: this is a simplification: this correlation depends on the location of DNA methylation. While promoter methylation may be repressing, methylation of gene body is positively correlated with gene expression levels (and H3K36 methylation)
  • Line 67: DNMT2 is not a DNA methyltransferase but a tRNA methyltransferase in mammals and should not be included in the context of setting and maintenance of DNA methylation patterns
  • Line 73: DNMT3L belongs to DNMT3 family and shares sequence/fold similarity with DNMT3a and 3b but is catalytically inactive which is a key difference. It has several regulatory functions beyond increasing SAM binding (it stimulates the catalytic activity of DNMT3a/3b, is involved in targeting and chromatin recognition etc)
  • Line 117-118: it is unclear what the authors mean by “DNA methylation could be used as a marker to evaluate memory differentiation and commitment”. The concept should be expanded and explained better.
  • Line 121: unclear what the 1,144 DMRs refer to (what cells were compared in the original study)?
  • Line 123: “these regions included genes” suggest that the DMRs were larger than the genes, which is unlikely. Do the authors mean the DMRs were associated/linked to with genes….?
  • Lines 136-139 and 144-148: these parts are unclear, what is the mechanism behind? What is the exact contribution of DNA methylation? More precise/detailed mechanism should be provided to illustrate the concepts, otherwise the only thing the reader learns is that DNA methylation is associated with a regulation of a given gene.
  • Line 166: what do the authors mean by “presented a methylation sensitive activity”? this should be explained better.
  • Line 122: “increasing the unmethylated regions is secondary memory CD8 cells” is unclear and should be explained
  • Line 245: how is the INF gamma regulated?
  • Lines 257-263: is the hypomethylation of these genes linked to their increased expressions? How do they link to the biological function of TEM cells?

  • The review has no single figure, it would be useful to include some to illustrate the mentioned concepts or mechanisms

  • It would be useful to explain in the introduction how DNA methylation may regulate gene expression (by which mechanisms) to illustrate the concepts. Along the same lines, it would be useful to establish in the introduction that DNA methylation patters are cell type specific and contribute to cellular differentiation and fate specialisation, as this argument is used often in the manuscript.

  • It may be useful to mention the 3 TET family members in the intro, as TET 2 is discussed more in details later in the manuscript.

  • Language:
  • Line 19: “catalyzed” remove d (catalyze)
  • Line 41: “by adding methyl to cytosine residues” -> groups should be added following methyl
  • Line 43: it is unclear what “their” refers to (histone tails?)
  • Line 77: remove “more recently” as it is confusing
  • Line 95: “phenotypes” is unclear here, do the authors mean “cells displaying these phenotypes”?
  • Line 96: this is a very vague line (different cells, several activities), more precise formulation would be helpful here
  • Line 99: “accompanied with” – should be changed to “accompanied by”
  • Line 120: add “analysis” after “genome-wide DNA methylation”
  • Line 147: “of” is missing after “control”
  • Table1: Correct DnmT3a to Dnmt3a and add references to the respective studies
